# Pre-fusion RSV F strongly boosts pre-fusion specific neutralizing responses in cattle pre-exposed to bovine RSV

Ann-Muriel Steff[1], James Monroe[1,4], Kristian Friedrich[1], Sumana Chandramouli[1], Thi Lien-Anh Nguyen[2], Sai Tian[1], Sarah Vandepaer[3], Jean-François Toussaint[2] & Andrea Carfi[1,5]

Human respiratory syncytial virus (hRSV) is responsible for serious lower respiratory tract disease in infants and in older adults, and remains an important vaccine need. RSV fusion (F) glycoprotein is a key target for neutralizing antibodies. RSV F stabilized in its pre-fusion conformation (DS-Cav1 F) induces high neutralizing antibody titers in naïve animals, but it remains unknown to what extent pre-fusion F can boost pre-existing neutralizing responses in RSV seropositive adults. We here assess DS-Cav1 F immunogenicity in seropositive cattle pre-exposed to bovine RSV, a virus closely related to hRSV. A single immunization with non-adjuvanted DS-Cav1 F strongly boosts RSV neutralizing responses, directed towards pre-fusion F-specific epitopes, whereas a post-fusion F is unable to do so. Vaccination with pre-fusion F thus represents a promising strategy for maternal immunization and for other RSV vaccine target populations such as older adults.

[1] GSK Vaccines, 14200 Shady Grove Road, Rockville, MD 20850, USA. [2] GSK Vaccines, Rue de l'Institut 89, Rixensart 1330, Belgium. [3] Keyrus Biopharma, Chaussée de Louvain 88, Lasne B-1380, Belgium. [4] Present address: Takeda Vaccines, 75 Sidney Street, Cambridge, MA 02139, USA. [5] Present address: Valera LLC, 500 Technology Square, Cambridge, MA 02139, USA. Correspondence and requests for materials should be addressed to A.-M.S. (email: ann-muriel.x.steff@gsk.com) or to A.C. (email: andrea.carfi@valeratx.com)

Human respiratory syncytial virus (hRSV) is the major cause of bronchiolitis and pneumonia in infants and is responsible for about 50% of all hospitalizations caused by respiratory infections in children between 0 and 2 years of age[1, 2]. It has been estimated that hRSV infects virtually all children by the age of 2 and peak hospitalization occurs between months 2 and 4[3]. There is no specific antiviral treatment recommended for hRSV infection and the only currently available prophylactic is a monoclonal antibody, Palivizumab (Synagis), used to prevent disease in the highest risk infants[4]. Most people get re-infected by hRSV repeatedly during their lifetime and infection has been shown to cause substantial morbidity and mortality among the elderly[5], leading each year on average in the United States to 177,000 hospitalizations and 14,000 deaths among adults 65 years old and above[6].

Maternal hRSV neutralizing antibodies transferred to the fetus through the placenta during pregnancy confer some level of protection during the first 1-2 months of life[7–10]. However, as these passively transferred antibodies wane, babies become more susceptible to hRSV infection[11]. One strategy to increase and extend protection during the first 4-6 months of life, the most critical for severe hRSV infections, is to vaccinate pregnant women during the third trimester of pregnancy, effectively boosting the pre-existing hRSV immune response and increasing neutralizing antibody titers in the newborn[12, 13].

The RSV fusion glycoprotein (F) is a conserved target of neutralizing antibodies[14], including Palivizumab and the closely related monoclonal antibody, Motavizumab[15]. Therefore, F is a promising antigen for RSV candidate vaccines. F is a class I viral fusion protein that mediates membrane fusion during viral entry. The F protein is in a metastable state on the viral envelope and undergoes a dramatic conformational change from a pre-fusion to a post-fusion state during virus entry, first described for the related parainfluenza (PIV) fusion proteins[16, 17]. Conformational changes in F allow viral and host membranes to come into close proximity and to fuse.

In the pre-fusion conformation, the heptad repeat A (HRA) region is associated with the globular head while in the post-fusion conformation HRA has extended from the head and the heptad repeat B (HRB) region has rearranged to associate with

the HRA region, forming a very stable 6-helix bundle. Recent crystallographic studies have defined the structures of RSV F in the pre- and post-fusion states to atomic resolution[18–20]. Moreover, researchers in other laboratories have succeeded in generating RSV F molecules, such as PreF-GCN4, DS-Cav1 and SC-TM that are stabilized in the pre-fusion conformation by introducing mutations that prevent rearrangement of HRA and by adding a trimerization sequence at the C-terminal end of HRB[21, 22–24].

Structural and biophysical studies coupled to immunization experiments have helped in defining the location of neutralizing sites on hRSV F and the importance of the stability of quaternary epitopes for raising high titers of neutralizing antibodies[24]. Among the sites common to both pre- and post-fusion F are Site II, binding the antibodies Motavizumab and 47F and site IV, binding 101F. Site Ø is only present on pre-fusion F and is recognized by the D25 antibody. Another pre-fusion-specific antibody, MPE8, has been shown to recognize an epitope that is conserved across four related paramyxoviruses, hRSV, bovine RSV (bRSV), human metaneumovirus and pneumonia virus of mice[25, 26]. A unique trimer-specific neutralizing antibody, AM14 has also been described[27, 28]. Finally, several human neutralizing antibodies isolated from memory B-cells of infected subjects have been recently reported[29]. Importantly, antibody depletion studies have revealed that the majority of hRSV neutralizing antibodies in sera from infected subjects target pre-fusion F while post-fusion F depletes only a small fraction of the total sera neutralization activity[30].

Immunization of RSV-naïve mice has demonstrated that pre-fusion F raises higher titers of neutralizing antibodies than post-fusion F[22, 23, 31]. Stabilized DS-Cav1, when combined with adjuvants, has been shown to raise between 8- and 15-fold higher hRSV neutralizing antibody titers than post-fusion F in mice and cotton rats, and up to 80-fold in non-human primates[22]. More recently, the superior immunogenicity of stabilized pre-fusion bRSV F proteins compared to post-fusion has been demonstrated in young RSV-naïve calves[32]. It remains unknown, however, whether stabilized pre-fusion F molecules can boost neutralizing responses in a primed population, and if so, what level of boosting could be expected. RSV priming can be achieved in rodents such

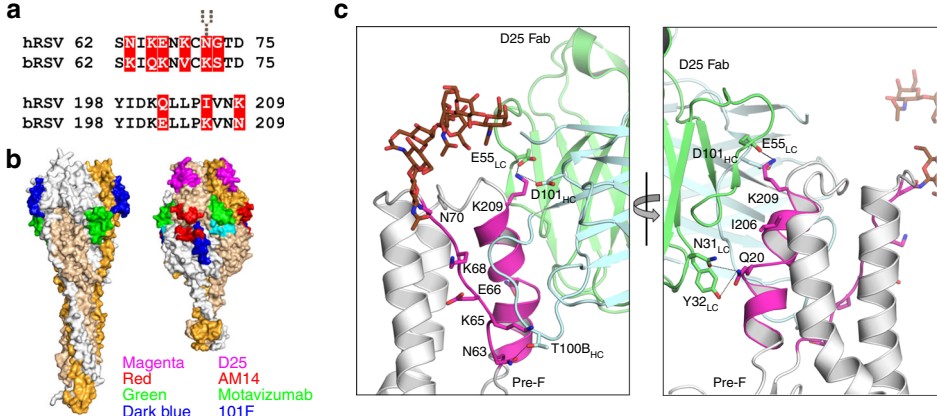

**Fig. 1** Neutralizing epitopes of human and bovine RSV F proteins. **a** Sequence alignment of Site Ø residues with differences between hRSV and bRSV F highlighted in red. The glycosylation site at N70 of hRSV F is indicated with a branched carbohydrate symbol. **b** Neutralizing epitopes of hRSV F mapped on the surface of post-fusion (left) and pre-fusion (right) models of the bovine counterpart. Individual monomers in each trimer are colored white, tan and orange. Epitopes depicted are for the following monoclonal antibodies: Motavizumab (Site II, green), 101F (Site IV, dark blue), D25 (Site Ø, magenta), MPE8 (cyan) and AM14 (red). **c** Molecular detail of Site Ø (magenta) in human RSV F with residues that differ from bRSV F and the corresponding key contact residues in D25 (light blue and green) shown as sticks. Dotted lines represent hydrogen bonds between antigen and antibody. HC heavy chain, LC light chain

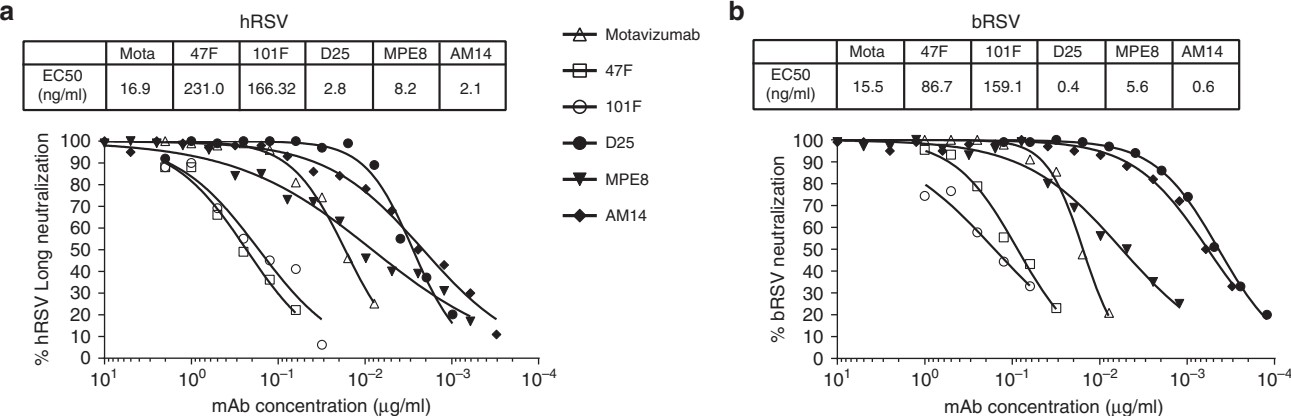

**Fig. 2** Neutralizing activity of different monoclonal antibodies targeting the RSV F protein. **a** hRSV; (**b**) bRSV. Effective concentrations (EC50) for each of the antibodies are indicated above the graphs (in ng/ml). Mota: motavizumab

as mice or cotton rats through hRSV intranasal inoculation. However, RSV poorly replicates in these species and relatively high amounts of virus are needed. This is in contrast to human infection, which occurs repeatedly throughout life, where hRSV infects the upper respiratory tract at low levels and replicates locally before infecting lower respiratory airways[33].

To circumvent the limitations of artificial RSV priming in rodents, we have exploited a primed cow model. Cattle are naturally infected by bRSV, a virus genetically related to hRSV, causing lower respiratory tract disease in young animals[34, 35]. We show here that, in this primed animal model, a single administration of non-adjuvanted pre-fusion F DS-Cav1 markedly boosts pre-existing RSV neutralizing responses and pre-fusion specific neutralizing antibodies, while post-fusion F does not. Therefore, pre-fusion F represents a promising candidate for further increasing RSV neutralizing antibodies in pregnant women and for other vaccine target populations, such as the elderly.

## Results

**hRSV pre-fusion-specific antibodies neutralize bRSV.** Bovine RSV F shares 90% sequence identity with hRSV F. However, sequence alignments show that several mutations cluster in the Site Ø (Fig. 1a), perhaps as a result of immune pressure. To assess the effect of mutations on epitopes recognized by hRSV F-neutralizing antibodies (Fig. 1b), we generated a homology model of pre-fusion bRSV F based on the crystal structure of pre-F DS-Cav1. We then mapped on this model the epitopes recognized by monoclonal antibodies 101F[36], Motavizumab[15], D25[19], MPE8[25] and AM14[28] and amino acids, within these epitopes, that have exposed side chains and differ between h- and bRSV F. This analysis revealed that the Motavizumab, 101F, MPE8 and AM14 binding sites are identical in the two molecules and that only two amino acids among those with side chains contacting D25 (Fig. 1c) differ between human and bovine RSV F (N63K and Q202E). The other main difference between the two RSV F molecules is the presence of an N-linked glycosylation site at N70 in hRSVF that is absent in the bovine counterpart for the strain used in this study[37] (Figs. 1a, c). It should be noted however that the amino acid at position 70 is an asparagine in some bRSV strains[32].

The conservation of the neutralizing sites suggests that hRSV F antibodies could bind bRSV F and neutralize the virus. To confirm this hypothesis, we tested Motavizumab, 47F, 101F, MPE8, AM14 and D25 in a neutralization assay and confirmed that all of them neutralize bRSV infection (Fig. 2). Motavizumab,

101F and MPE8 retained similar neutralization potency against bovine and human RSV with Motavizumab and MPE8 being 10-fold more potent than 101F. Interestingly, AM14 and D25 were 4- and 7-fold more potent, respectively, in neutralizing bRSV compared to hRSV. Together these data demonstrate that bRSV F has antigenic and functional properties similar to hRSV F.

**Pre-fusion F boosts the pre-existing bRSV response.** The antigenic similarities between bRSV and hRSV F suggest that cattle could be a relevant animal model to test the boosting potential of RSV F proteins in a primed population. Animals were randomized, based on age and pre-existing bRSV antibody levels, into 3 groups. Two groups of 9 animals each were immunized with non-adjuvanted DS-Cav1 at two doses (60 and 400 µg) while a third group of 4 animals was immunized with non-adjuvanted post-fusion F (Post F) at the 400 µg dose. The Post F and pre-fusion DS-Cav1 used in this study were fully characterized biochemically, antigenically and also in an *in vivo* mouse immunogenicity experiment (Supplementary Figs. 1 and 2). Both proteins were shown to be of high quality and having the same in vitro and in vivo properties as reported in previous studies[18, 22].

Analysis of hRSV neutralization titers in individual sera revealed that all but one animal responded to a single DS-Cav1 immunization (Supplementary Fig. 3). On day 14, both groups immunized with DS-Cav1 at high or low dose showed significantly higher Geometric Mean Titers (GMTs) than animals immunized with Post F but there was no statistically significant difference between the high and low dose DS-Cav1 groups. In addition, similar trends were observed when comparing groups over the entire time range (Fig. 3a): a superiority of DS-Cav1 compared to Post F and no evidence of a difference between both DS-Cav1 regimens (60 and 400 µg).

When further characterizing the treatment effect, a statistically significant increase in neutralization titers from Day 0 to Day 14 ($P < 0.0001$, ANOVA model t-test) was detected in DS-Cav1 groups (Fig. 3a, Supplementary Fig. 3 and Supplementary Table 1). The Day 14/Day 0 Geometric Mean Ratios (GMR) estimates were 13.88 (CI: 8.47-22.73) and 7.34 (CI: 4.61-11.69) for the high and low dose DS-Cav1 groups, respectively while the one of Post F was 1.03 (CI: 0.51-2.06) (Fig. 3a and Supplementary Table 1).

For both DS-Cav1 groups, the peak of the response generally occurred on Day 14, with an average 2-fold decrease in neutralization titers over time from day 14 through day 56 (Fig. 3a). Overall, the calculated half-lives of hRSV neutralizing

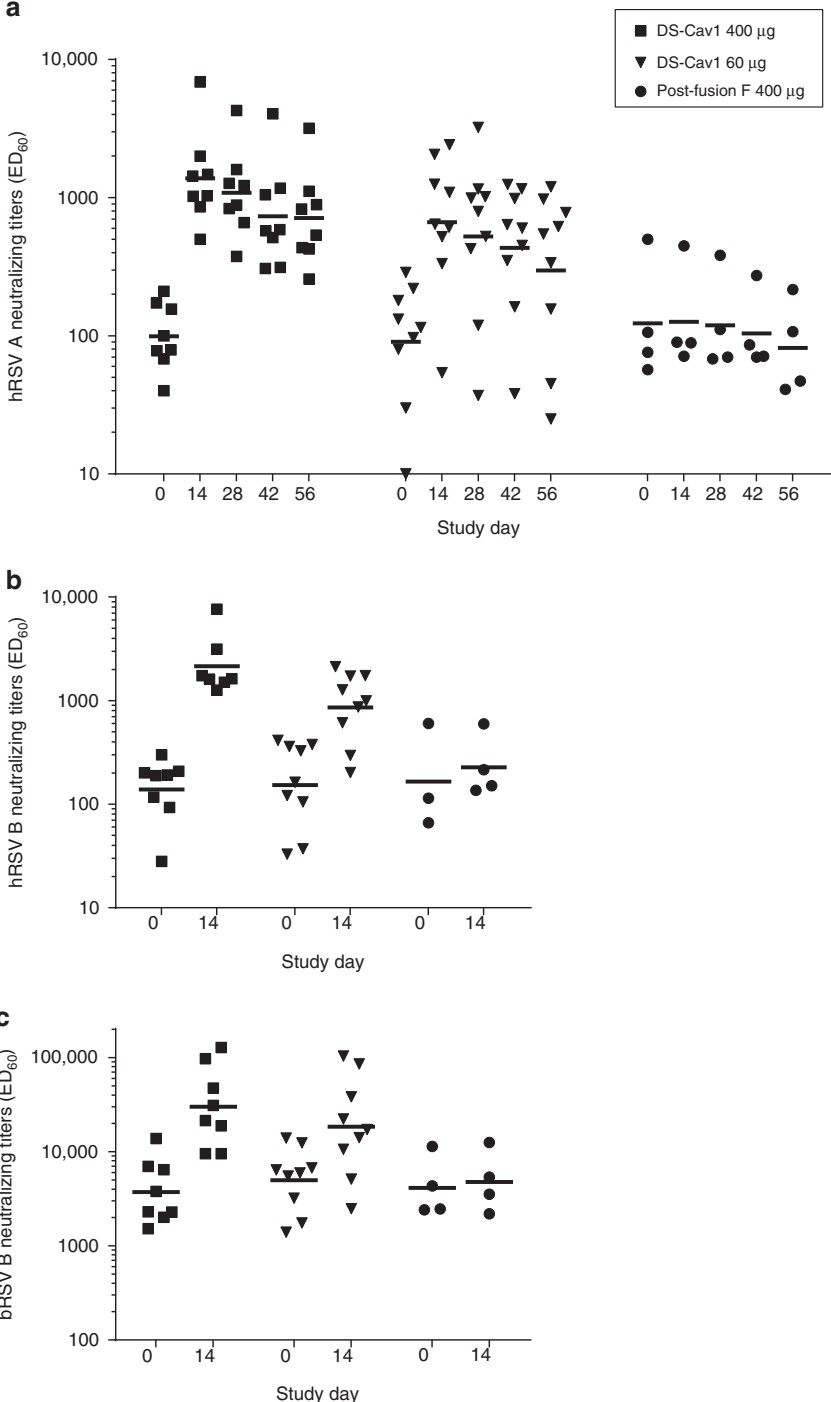

**Fig. 3** hRSV A (**a**), hRSV B (**b**) and bRSV (**c**) neutralizing antibody titers after vaccination with a single dose of non-adjuvanted proteins. Horizontal bars indicate the geometric mean titer (GMT) at each time point. One animal in the high dose DS-Cav1 group did not respond to the vaccine and was excluded from the graphs

antibodies were 46 and 37 days for the 400 and 60 μg doses of DS-Cav1, respectively.

Cattle sera were also tested for hRSV B and bRSV neutralization. Overall, hRSV B neutralization response was very similar to that observed with hRSV A (Fig. 3b and Supplementary Table 1), showing that antibodies raised through vaccination of bRSV-positive animals with a pre-fusion F derived from a hRSV A strain were cross-reactive with another hRSV group. A minimal increase in hRSV B neutralization response was observed after vaccination with PostF. bRSV neutralization titers were overall

higher compared to hRSV (Fig. 3c and Supplementary Table 1). Similarly to hRSV neutralization, GMTs were boosted by immunization with DS-Cav1 but not Post F. GMRs from Day 0 to Day 14 in serum from animals vaccinated with DS-Cav1 were slightly lower than for hRSV (GMR$_{DS-Cav1 \ high\_dose}$ = 8.06; CI: 3.40-19.1 and GMR$_{DS-Cav1 \ low\_dose}$ = 3.70; CI: 1.64-8.35). Together these data clearly demonstrate that a single dose of non-adjuvanted DS-Cav1 was able to boost pre-existing bRSV neutralizing antibody responses cross-reactive with hRSV whereas Post F was not. The superiority of DS-Cav1 was on the

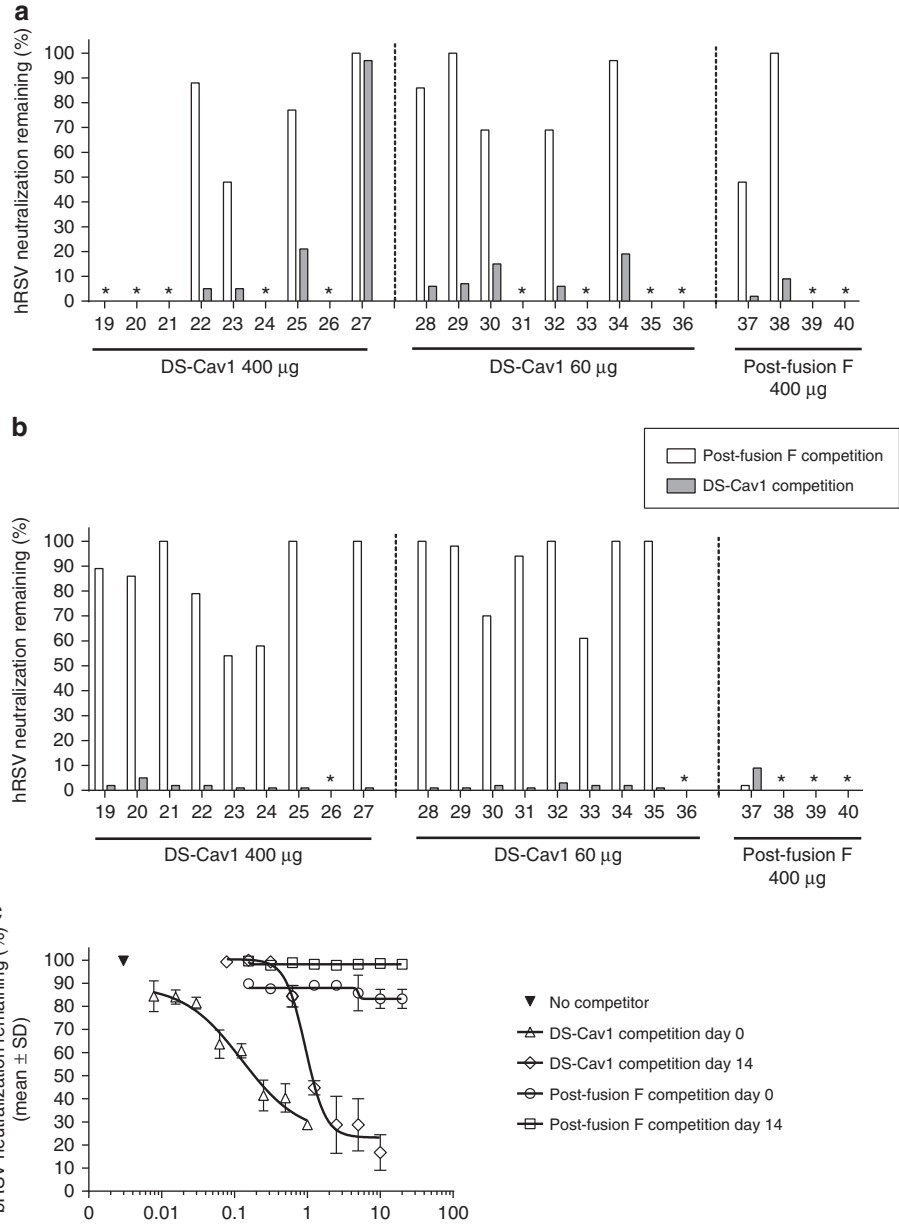

**Fig. 4** Remaining neutralization activity after serum pre-incubation with pre-F DS-Cav1 or post-fusion F. Day 0 (**a**) and Day 14 (**b**) individual serum samples from each of the treatment groups with neutralization titers > 100 were pre-incubated with either DS-Cav1 or post-fusion F (40 µg/ml), their hRSV neutralization titers were determined and the % remaining neutralization activity was calculated relative to neutralization titers without protein pre-incubation. Asterisks indicate animals for which neutralization titers were ≤ 100. **c** Day 0 or Day 14 serum pools from each of the treatment groups were pre-incubated with varying concentrations of DS-Cav1 or post-fusion F (competitor) and the % remaining bRSV neutralization activity was calculated as described above. Symbols indicate the mean and error bars the SD of duplicate measurements

boosting of functional antibodies since increases in bRSV F binding antibodies could be observed with both DS-Cav1 and Post F (Supplementary Table 2).

**Pre-fusion F mainly boosts pre-fusion-specific antibodies.** Having shown that bRSV and hRSV F have similar antigenic properties prompted us to assess which F conformer is targeted by neutralizing antibodies in sera from bRSV infected cattle and what antibody specificity was boosted by immunization with F. With this goal, we initially tested the ability of pre- or post-fusion F to deplete the RSV neutralizing activity in serum of cattle at day

0 and day 14 post-vaccination. For the day 0 samples only sera from animals with hRSV neutralization titers above 100 were analyzed.

The serum depletion assay revealed that in the majority of animals (10 out of the 11 cattle with measurable RSV neutralization titers), antibodies targeting pre-fusion F were responsible for most (≥ 80%) of the sera neutralization activity on day 0 (Fig. 4a). On the other hand, Post F depleted between 10–35% of total neutralization in 5 cows and up to 45–65% in 2 cows (23 and 37) (Fig. 4a). In 4 animals, minimal (< 10%), if any, neutralization inhibition could be observed when serum was pre-incubated with Post F. Finally, for animal No. 27 neither pre- nor

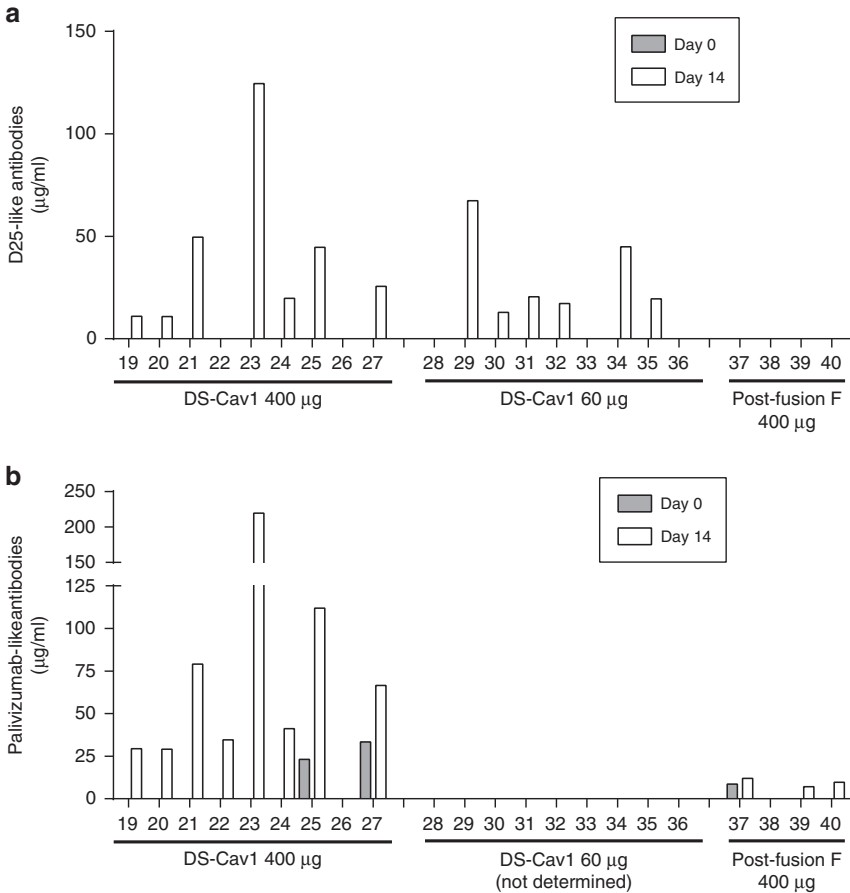

**Fig. 5** Antibody competition assays. Serial dilutions of Day 0 and Day 14 individual serum samples were pre-incubated with biotinylated D25 (**a**) or Palivizumab (**b**). D25- or Palivizumab-competing antibodies in the sample prevent binding of the biotinylated antibodies to the DS-Cav1 coated-plate, allowing the quantification of D25-like or Palivizumab-like antibodies. **a** D25-like antibodies: no D25-like antibodies were detected in any of the animals on Day 0. Asterisks indicate animals where D25-like antibodies were below the limit of detection either on Day 0 or Day 14. **b** Palivizumab-like antibodies: samples from the DS-Cav1 60 μg group were not tested. In the DS-Cav1 and Post-fusion F groups, absence of grey histograms indicates samples with palivizumab-like antibody levels below the limit of detection on Day 0

post-fusion F could inhibit the hRSV neutralization activity in serum (Fig. 4a). Together these data suggest that in naturally bRSV infected cattle, most neutralizing antibodies target pre-fusion F specific sites, though in some animals the response is more mixed and neutralizing sites common to both pre- and post-fusion F are recognized.

We then analyzed Day 14 sera from animals immunized with DS-Cav1 or Post F and found that DS-Cav1 was able to inhibit most of the hRSV neutralization activity in all vaccine groups (Fig. 4b), in contrast with depletion with Post F where neutralizing activity was minimally or very partially depleted in DS-Cav1-vaccinated animals. Post F was nonetheless able to deplete most of the neutralizing activity in the single animal vaccinated with Post F for which neutralizing antibody titers were high enough to perform the assay. Together these data demonstrate that DS-Cav1 pre-fusion F effectively boosted neutralizing antibodies targeting pre-fusion specific F epitopes.

Titration of DS-Cav1 into pooled sera from 8 animals with the highest bRSV neutralization titers at day 14 showed that 10-fold more protein was needed to obtain the same level of inhibition of bRSV neutralization than the corresponding pooled sera at day 0 (Fig. 4c). This result reveals a correlation between neutralization titers and pre-fusion F-specific antibody concentrations. Titration of post-fusion F into this serum pool had little effect in inhibiting bRSV neutralization (Fig. 4c).

**Pre-fusion F boosts D25 and Palivizumab competing antibodies.** Finally, we tested whether immunization with either DS-Cav1 or post-fusion F boosted D25- and Palivizumab-like antibodies in the sera of immunized animals using an ELISA competition assay. D25-like antibodies were below limit of detection prior to immunization (Day 0) in all animals, likely due to a low assay sensitivity, but were boosted and became quantifiable at various levels in most animals vaccinated with DS-Cav1 (Fig. 5a and Supplementary Fig. 4), with the exception of 4 animals (including animal No. 26 that did not respond to vaccination with DS-Cav1). In contrast, animals vaccinated with post-fusion F did not show any increase in D25-like antibodies, i.e. remained negative (Fig. 5a). This suggests that only DS-Cav1 was able to boost site Ø-specific antibodies even in the presence of pre-existing pre-fusion specific antibodies, and is consistent with the results of the neutralization competition experiments.

We also assessed the level of Palivizumab competing antibodies (PCA) raised by DS-Cav1 and post-fusion F in an ELISA competition assay. Only 3 of the 13 tested animals had quantifiable PCA concentrations, at low levels, on Day 0. All animals vaccinated with the high dose DS-Cav1 showed a marked increase in PCA concentrations (Fig. 5b) 14 days after immunization and PCA became detectable, although at low levels, in 3 out of 4 animals vaccinated with Post F (Fig. 5b). Together these data indicate that pre-fusion F is also able to raise

PCA to levels even higher than post-fusion F. It should be noted, however, that the nature of antibodies induced by vaccination with either pre-fusion or post-fusion F and affecting binding to the palivizumab site, may differ.

## Discussion

RSV infection is the major viral cause of respiratory disease in infants during the first months of life and drives considerable morbidity and mortality in the elderly population. Despite more than 40 years of vaccine research, there is no licensed vaccine available against RSV. Development of a vaccine was stymied in the 1960's when a formalin-inactivated RSV vaccine candidate made subsequent RSV disease more severe[38].

A maternal immunization strategy has been proposed as a way to provide protection while overcoming some of the risks associated with active immunization of newborns with a RSV subunit-based vaccine. Titers of maternally-derived neutralizing antibodies demonstrate an inverse association with the incidence of RSV acute lower respiratory tract disease during the first 6 months of life. A decrease in neutralizing titers below 7.5 log2 ($\approx$ 1:180) was associated with an increase in the number of hospitalizations in infants younger than 6 months of age. It has been estimated that every 2-fold rise in cord blood neutralizing antibody titers reduces the risk of RSV-associated hospitalization in the first 6 months of life by 26–30%[11, 39]. Importantly, the success of the maternal immunization approach is dependent on the level of boosting of the pre-existing neutralizing antibodies achieved by immunization of mothers, trans-placental transfer of antibodies, and the half-life of antibodies in the newborns.

Assessing what fold increase in RSV neutralization titers can be achieved in primed species has been difficult because rodents such as cotton rats and mice are only poorly infected by hRSV. Here, we used cattle as a primed animal model to study the effect on RSV neutralizing responses obtained by immunization with a pre-fusion hRSV F vaccine candidate and compared it to a post-fusion F molecule. Cattle are naturally infected by bRSV and at young age develop a disease that has similarities to that induced by hRSV in humans. In the current study, all but two of the recruited animals were "primed" for bRSV, likely resulting from natural infections throughout life.

We first confirmed that bovine and human RSV F molecules are antigenically similar. Sequence and structural considerations revealed that Sites I, IV and the MPE8 and AM14 quaternary binding sites are strictly conserved among bovine and human RSV F, while only two residues among those contacting D25 (Site Ø) differ. We also showed that both viruses are neutralized by a panel of representative antibodies targeting the above neutralization sites. Interestingly, Motavizumab and 101F, two antibodies that bind to both pre- and post-fusion F, and the pre-fusion specific antibody MPE8 had similar neutralization potency against bovine and human RSV. In contrast, the pre-fusion specific D25 antibody was 7-fold more potent in bRSV neutralization. We speculate that differences in affinity of D25 for bRSV and hRSV F, or the presence of a glycosylation site in the vicinity of the D25 epitope in hRSV F, might be the cause. However, we cannot exclude that other factors, such as the flexibility of this region or different antibody-antigen interactions in the two viral F proteins, may play a role.

Analysis of sera from cows naturally infected by bRSV confirmed the cross-neutralization potential of bovine antibodies towards hRSV A and hRSV B, as expected from the homology between bovine and human RSV F proteins. Neutralization competition assays revealed that most of the hRSV neutralizing response in bovine serum could be depleted by incubation with DS-Cav1, while post-fusion F only depleted between 10 and 30%

of neutralization in some cattle and was ineffective in others. Previous studies have demonstrated that the majority of the hRSV neutralizing antibodies in sera from RSV seropositive human subjects target the F protein in the pre-fusion conformation[30]. In these studies, post-fusion F depleted only 10-30% of the neutralizing fraction, whereas the DS-Cav1 depleted between 70 and 90% of neutralization when added to the sera. Therefore, the bRSV neutralization response in cattle mimics the situation in primed humans and targets mainly pre-fusion F specific antigenic sites. We note that in one cow, the hRSV neutralization response before immunization was not affected by the competition with the F proteins. This result suggests that the neutralizing antibody response might have been raised against a different antigen, possibly the RSV G protein[40]. Finally, comparison of the neutralization potency of sera from bRSV-positive cattle against bovine and human RSV revealed systematically higher potency against the bovine virus. The reasons for the difference in neutralization of bRSV vs hRSV by cattle serum are presently unknown but are reminiscent of the higher neutralization potency of the D25 antibody for bRSV or could alternatively be due to differences in antigenicity between the corresponding F proteins.

A single immunization of bRSV seropositive cattle with non-adjuvanted DS-Cav1 resulted in a statistically significant increase in hRSV A and hRSV B neutralizing titers from Day 0 to Day 14. This is in contrast with primed rodent models where a single administration of a non-adjuvanted F protein has little boosting effect on pre-existing neutralizing antibody titers, likely due to the poor replication of hRSV in this animal species. In this respect, the primed bovine model seems more reliable to predict the boosting potential of non-adjuvanted protein RSV vaccines in humans[41, 42]. The boost in the neutralization responses was higher for the high dose group (14 vs 7-fold in the high and low dose groups, respectively), although this difference did not reach statistical significance, peaked at day 14 and decreased at the subsequent data points. In addition, one animal (animal No. 36), that did not have detectable hRSV F neutralizing antibodies at Day 0, had a 5-fold increase in neutralization titers after immunization with pre-F DS-Cav1, suggesting that even very low pre-fusion specific antibodies can effectively be boosted by non-adjuvanted pre-fusion F. Analysis of the same cattle sera on bRSV neutralization revealed similar results as observed with hRSV, although the GMR between Day 14 and Day 0 were somewhat lower. This discrepancy may be due to the use of the hRSV pre-fusion F as vaccine, which may have slight differences in antigenic properties compared to bRSV pre-fusion F, thus boosting a subset of antibodies that would not neutralize bRSV. This finding also suggests that immunization with hRSV pre-fusion F may achieve an even higher boosting of neutralizing antibodies in the human population, compared to what reported here, due to the homologous priming (i.e with hRSV). Most importantly, DS-Cav1 boosted neutralizing responses even from high starting titers on Day 0, which has been a limitation in some of the protein-based RSV vaccines tested so far[43], suggesting that vaccination with DS-Cav1 could be effective in boosting neutralizing responses in subjects with a recent RSV infection.

The antibody half-lives calculated in this study, 46 and 37 days for the 400 and 60 µg doses of DS-Cav1, respectively, are consistent with studies reporting RSV neutralizing antibody half-lives of 36 to 38 days in newborns following RSV infection[44, 45] or vaccination[46] of their mothers. Other studies however have shown much longer antibody persistence in newborns[9] or vaccinated children[47]. We tried to estimate the duration of protection that would be conferred by a 14-fold increase in neutralization titers from baseline, as seen 14 days after vaccination with high dose DS-Cav1. For this calculation we assumed

that pregnant women would have baseline neutralizing antibody titers of 1:300[41, 42], that the trans-placental transfer ratio would be 1:1 and that the half-life of antibodies would be 30 days[48]. Based on this rather conservative scenario, a 14-fold increase in baseline titers would result in 4-5-months of neutralizing antibody titers above the protective titer of 1:180 in babies. Of note, a direct "maternal immunization" study, whereby cows would be immunized during pregnancy and protection from RSV challenge would be evaluated in calves, is not feasible due to the antibody-impermeable structure of bovine placenta[49].

Immunization with post-fusion F did not affect the pre-existing RSV neutralization titers (GMR of 1). Consistent with this result, it has recently been shown that immunization of human subjects with an RSV vaccine consisting of non-adjuvanted nano-particles composed of full length hRSV F protein in the post-fusion conformation boosted neutralization titers by a maximum 2.3-fold after a single dose[41]. We also note that the contribution of site II and site IV epitopes, shared by pre- and post-fusion F, to the overall neutralizing antibody response is probably minor, as addition of post-fusion F to the sera only marginally affected neutralizing titers.

It could not be ruled out in this study that some animals may have received a bovine RSV vaccine previously. An intra-nasal live-attenuated vaccine (Rispoval®) is the most commonly administered vaccine in the country of origin of the animals (Belgium) and does not confer sterilizing immunity[50]. It is generally admitted that vaccination with attenuated RSV vaccines mimic the immune response triggered by natural infection. In case an inactivated vaccine would have been used, the antibody response after vaccination could have differed due to a different priming status, as inactivation of RSV may lead to changes in the conformation of the fusion protein, with PostF predominating after inactivation[51]. However, this change in F protein conformation has only been demonstrated for the human formalin-inactivated RSV vaccine and not for inactivated bovine RSV vaccines. Of note, our neutralization competition experiments revealed that prior to any immunization, neutralizing antibodies in bovine serum were mainly directed against pre-fusion rather than post-fusion F, as in RSV infected humans. We therefore hypothesize that the immune response induced by vaccination with either DS-Cav1 or PostF would not overly differ whether the pre-existing immune response was driven either by natural infection or vaccination, making cattle a good animal model to address the potential of an RSV vaccine to boost pre-existing RSV neutralizing antibodies.

In conclusion, our data support the concept of using a pre-fusion F-based vaccine as a way to boost hRSV neutralizing antibodies in individuals with pre-existing immunity. Translated to maternal immunization, the use of pre-fusion F is expected to extend the duration of protection conferred by maternally transferred antibodies, thereby lowering RSV burden of disease in infants.

## Methods

**Cattle immunization**. Forty cows were screened and randomized into 4 experimental groups of 9 animals, 2 of these groups being the DS-Cav1 vaccine groups and 2 other groups unrelated to the present manuscript (2 to 7 years-old). The 4 remaining calves were allocated to the post-fusion F vaccine group (4 to 10 years-old). Randomization was performed with the SAS procedure "PROC OPTEX", optimizing a design based on a covariate model containing "Age" and "Serology status" variables. Serology status was based on the level of pre-existing bRSV Immunoglobulin G (IgG), as determined in a semi-quantitative Enzyme-Linked Immuno-Sorbent Assay (ELISA). A power analysis was used to determine the minimum number of animals required to detect a 4-fold difference between groups with a 95% confidence and 80% power, assuming an SD of 0.4 $\log_{10}$. The study was not blinded.

Animals received a single intramuscular (IM) immunization of 2 ml in the left neck of either non-adjuvanted DS-Cav1 at 60 or 400 µg doses, or of non-

adjuvanted post-fusion F at the 400 µg dose. Doses were adjusted compared to what was previously used in small animals and non-human primates (NHP) to take into account the larger size of cattle. Blood was collected on days 0, 14, 28, 42 and 56 post-immunization for serum preparation and serological analyses. Immunogenicity of the different vaccines was tested in a series of assays, namely hRSV and bRSV neutralization assays, neutralization inhibition assays and D25- and palivizumab-competition assays. One animal did not seroconvert in all these assays and was considered an outlier. This animal (No. 26) was excluded from GMT or GMR calculations. Individual results for all animals included in this manuscript are presented in Supplementary Table 1.

The in life part of the study was conducted at the Centre d'Économie Rurale (CER) Group (Marloie, Belgium), in accordance with Belgian and European laws, guidelines and policies for animal experimentation, housing and care, as per treaty ETS No.123, Belgian Royal Decree of 29 May 2013 and European Directive 2010/63/EU. The study was approved by the CER Group Ethical Review Committee under number CE/Santé/ET/012 (adopted in 2015). Animals were purchased from local farmers and group-housed (4-5 animals per pen of 40 m$^2$), at room temperature, under natural light, with straw bedding. Cows were fed ad libitum with hay available at all times. Each cow received about 2 kg pellets per day. Cows had ad libitum access to drinking water during the whole study. Animal health was monitored daily by trained staff. The overall clinical status was satisfactory during the whole study. The administration site was monitored daily for local reactions. No or little pain and very light swelling were observed on the day of vaccination.

**Mouse immunization**. Six to eight-week old Balb/c female mice (Charles River Laboratories, USA), N = 8/group, were immunized twice, 21 days apart, with either a 3 µg or a 0.3 µg dose of proteins, mixed with an oil-in-water adjuvant (MF59). Serum was collected 14 days after the second immunization and hRSV A neutralizing antibody titers were measured as described below. The experiment was approved and performed according to Novartis Institutional Animal Care and Use Committee guidelines.

**Protein expression, purification and characterization**. Cloning, expression and purification of post-fusion RSV-F have been previously described[18]. A Chinese Hamster Ovary (CHO-K1A) cell[52] stable pool expressing pre-fusion RSV F DS-Cav1 with a C-terminal thrombin-cleavable His-tag followed by a double Strep tag II was cloned based on published literature[22]. The protein was purified by passing harvest medium buffer exchanged into PBS over a Strep-tactin Superflow column (Qiagen) and eluted with elution buffer (100 mM Tris pH 8, 150 mM NaCl, 1 mM EDTA and 2.5 mM desthiobiotin). The protein was then subjected to thrombin (Sigma-Aldrich) cleavage overnight at 4 °C, followed by a polishing step of size exclusion chromatography. The final protein was dialyzed into 10 mM potassium phosphate pH 7.2 and concentrated to ~ 1 mg/ml. Protein integrity and trimeric state was confirmed by Ultra Performance Liquid Chromatography (UPLC) on a BEH200 Size Exclusion Chromatography column connected to a Waters Acquity UPLC system. Antibodies used in the study were obtained from different sources: palivizumab[4] (MEDI-493, Synagis, MedImmune, Inc, Gaithersburg, MD) is commercially available; MPE8 was obtained from Humabs Biomed[25]; R145[53] and 101F[36] were received from Dr. José Melero; 47F[54] was produced from a mouse hybridoma; D25[19], AM14[28] and motavizumab[15] were expressed in 293 Expi cells (Gibco, Cat. No. A14527) and purified over a protein A column and buffer exchanged into 25 mM Tris pH 7.5 and 150 mM NaCl.

**Antigenicity ELISA**. Ninety six well plates were coated with DS-Cav1 or Post F (0.1 µg/ well in phosphate buffered saline, PBS) overnight at 4 °C. Following overnight incubation, wells were washed 3 times with 300 µl/well of PBS containing 0.05% (w/v) Tween 20 (wash buffer). The wells were blocked with 1% bovine serum albumin (BSA) in PBS for 45 min at room temperature (RT) and then washed. Antibodies 47F, 101F, D25 and R145, which were previously biotin conjugated using a commercial kit (Thermo Scientific) were added to the plates in 3 fold serial dilutions in sample buffer (1% BSA and 0.1% (w/v) Triton X-100 in PBS) and incubated for 90 min at RT. Following a wash, horseradish-peroxidase conjugated avidin (Vector) diluted 1:10000 in sample buffer was added to the plates and incubated for 60 min at RT. After a final wash tetramethylbenzidyl substrate (Rockland) was added to the plates for 30 min at RT and the reaction was stopped with 2.0 N sulfuric acid. The optical density was determined spectrophotometrically at 450 nm wavelength using a microplate reader.

**Bovine RSV F ELISA**. A semi-quantitative bRSV F antibody detection ELISA kit (CER Group, Belgium, Cat. No. K.BRSVsero) was used to determine the amount of anti-bRSV F antibodies in serum. Half of the plates were coated with an anti-bRSV F monoclonal antibody (mAb) (CER Groupe, Belgium, Cat. No. IMM-COM-MON-002) and incubated with bRSV (strain RB94[50]) produced by virus culture on Vero monolayer cells (bRSV + ). The second half of the plate consisted of control wells (bRSV−) with mAb coating but no bRSV, to correct for serum non-specific binding. Serum diluted 1:100 was added to the plates and bound anti-bRSV antibodies were detected with a peroxidase labeled anti-bovine IgG conjugate and revealed with the enzyme substrate tetramethybenzidine. Plates were read at 450 nm and the signal of each bRSV− well subtracted from that of the corresponding

bRSV + well. In parallel to unknown serum samples, a positive control consisting of a reconstituted freeze-dried bRSV antibody was run on each plate and used to normalize sample results. According to the normalized value for each sample, results were expressed on a zero to ++++ scale.

**RSV neutralization assay**. Two-fold serial dilutions of heat-inactivated (HI) sera were pre-incubated with approximately 100 plaque forming units of hRSV A Long (ATCC, VR-26), hRSV B 18537 (ATCC VR-1580) or bRSV (ATCC, VR-1485) in phosphate-buffered saline (PBS) with 5% HI-Fetal Bovine Serum (FBS) for 2 h at 37 °C/5% $CO_2$ prior to inoculation of HEp-2 cells (ATCC, CCL-23) in 96-well plates. The inoculum was removed after 2 h and cells were overlaid with 0.75% methyl cellulose/Eagle's minimum essential medium (EMEM)/5% HI-FBS and incubated for 40–46 h (hRSV)/65–70 h (bRSV). Cells were then fixed with 10% neutral-buffered formalin and permeabilized with 0.5% saponin, and infectious foci were stained and visualized with a cocktail of mouse anti-RSV F and anti-nucleoprotein antibodies (AbD Serotec, MCA490 and MCA491G, 1:1000 dilution), followed by horseradish peroxidase (HRP)-conjugated goat anti-mouse IgG (Southern Biotech, 1032-05) and TrueBlue Peroxidase Substrate. Foci were counted using a CTL Immunospot S5 UV Analyzer. The neutralization titer was defined as the reciprocal of the highest interpolated serum dilution producing a 60% reduction in foci ($ED_{60}$), relative to virus-control wells without serum. If a sample's titer was below the first serum dilution tested, 1:20, it was assigned a titer of 10.

**Neutralization competition assay**. Serum samples were incubated for 1 h at room temperature with and without purified RSV recombinant proteins (DS-Cav1 and post-fusion RSV F), at a concentration of 40 µg/ml. These samples were then assayed in a RSV micro neutralization assay. Two-fold serial dilutions of the protein/serum mix were made in a 96-well assay plates. RSV virus was added to all wells and the plates were then run as per the neutralization assay protocol.

**Protein dilution curves**. Two-fold serial dilutions of purified RSV recombinant proteins (DS-Cav1 and post-fusion RSV F) were diluted in Dulbecco's Phosphate Buffered Saline (DPBS) supplemented with 5% HI-FBS in 96-well assay plates. Proteins were incubated with serum diluted 1:40 for 1 h at room temperature (RT). RSV virus was added to all wells and the plates were then run as per the neutralization assay protocol.

**Antibody competition ELISA**. A competition ELISA was performed to determine the concentration of Palivizumab- or D25-like antibodies in serum. Either Palivizumab or D25 antibody (referred to as "tracer") was biotin conjugated using a commercial kit according to manufacturer instructions (EZ-Link NHS-PEG4-Biotin, No-Weigh Format, Thermo Scientific). 96-well ELISA plates (Immuno F96 MaxiSorp, Nunc) were coated with 100 µl/well of RSV F DS-Cav-1 diluted to 2 µg/ml in Phosphate Buffered Saline (PBS). Following overnight incubation at 4 °C, wells were washed with PBS containing 0.05% (w/v) Tween 20 (wash buffer). The wells were blocked with 1% (w/v) bovine serum albumin (BSA) in PBS for 90 min at room temperature (RT). Bovine serum samples (starting 1:10) or unlabeled Palivizumab (standard, starting at 5 µg/ml) were serially diluted two-fold in PBS containing 1% (w/v) BSA and 0.1% (w/v) Triton X-100 (sample buffer). Sample and standard dilutions were combined in equal volumes with 20 ng/mL tracer. The ELISA plates were washed and tracer-sample/standard mixtures were transferred to the plates. Eight wells contained tracer only for determination of the tracer signal (tracer-only binding). Plates were incubated were then washed and incubated with HRP-conjugated avidin (Vector cat# A-2004) diluted 1:10000 in sample buffer at 100 µl/well for 1 h at RT, followed by a wash and incubation for 20 min with 100 µl/well of TMB substrate (Rockland cat# TMBE-1000) at RT. Following incubation, the reaction was stopped by adding 100 µl/well of 2.0 N Sulfuric Acid (BDH cat# BDH3500). The optical density was determined at 450 nm using a microplate reader (Infinite M200 NanoQuant, Tecan).

Percent inhibition of the tracer-only binding was calculated for each standard and sample dilution and plotted according to concentration (standards) or dilution (samples). For standards, the concentration of unlabeled D25 or Palivizumab leading to 50% inhibition of the corresponding tracer (EC50) was calculated in GraphPad Prism. For samples, the dilution corresponding to 50% inhibition was calculated in a similar manner, but only if the lowest sample dilution (1:10) was above 50% inhibition. This dilution was multiplied by the D25 or Palivizumab (standard) EC50 concentration to establish D25- or Palivizumab-like concentrations in each sample.

**Half-life calculations**. Half-lives of hRSV neutralizing antibodies were calculated by non-compartmental analysis of the GMT for each of the DS-Cav1 groups (400 and 60 µg) with PKSolver[55].

**Statistical analyses**. Individual animal results are presented for all serology assays. One animal that did not seroconvert in any of the serological tests performed was excluded from subsequent analyses. An ANOVA model for repeated measures including time, treatment and time by treatment as fixed factors was fitted on the log-transformed neutralizing antibody titers measured at all time points (day 0 to

day 56). The variance-covariance matrix of the residuals was assumed unstructured. Diagnostics (including residual plot, quantile-quantile plot and likelihood-based statistics) were computed to validate the model assumptions. The GMTs and GMRs of interest were derived from this model (with 95% confidence intervals). Comparisons for which the 95% confidence interval of the corresponding GMR did not encompass 1 were considered as significantly different. Statistical computations were performed using either the GraphPad Prism software or SAS 9.2.

For comparison of hRSV and bRSV neutralization potency of different mAbs, EC50 were calculated using non-linear regression of log(concentration) vs normalized % neutralization. For the determination of D25- and Palivizumab-like concentrations, the data was fitted using a 4-parameter logistic regression of log (dilution) vs % inhibition. The sample dilution corresponding to 50% inhibition was calculated.

**Data availability**. The data that support the findings of this study are available from the corresponding author on reasonable request.

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

## Acknowledgements

This work was sponsored by GlaxoSmithKline Biologicals SA which was involved in all stages of the study conduct and analysis. We thank Éric Fichant and Philippe Delahaut at CER Groupe for conducting the in life part of the study, Frédéric Renaud for statistical support and Sylvie Bertholet for critically reviewing the manuscript. We also thank Ulrike Krause for publication support.

## Author contributions

A.C., A.M.S. and J.F.T. designed the study. A.C., S.C., J.M., K.F., T.L.A.N., A.M.S. and S.T. were involved in methods selection and development. S.C., K.F., J.M., T.L.A.N. and S.T. acquired the data. A.C., S.C., K.F., J.M., T.L.A.N., A.M.S., J.F.T. and S.V. analyzed and interpreted the results. A.C., J.M. and A.M.S. wrote the manuscript. All authors were involved in revising the manuscript critically for important intellectual content. All authors had full access to the data and approved the manuscript before it was submitted by the corresponding authors.
