## [Peer Review File · Nature Communications]

Reviewers' comments:

Reviewer #1 (Remarks to the Author):

This manuscript describes the use of DS-Cav1 F vaccine, a prefusion RSV F vaccine, in RSV seropositive cattle to determine if DS-CavF1 can induce a significant rise in RSV-specific neutralizing antibody response that is comprised primarily of responses to the pre-fusion antigenic sites. DS-Cav1 F is a pre-fusion F molecule that has been stabilized by introducing mutations that prevent conformational change to the post-fusion F. Both a 60 ug and 400 ug dose were evaluated without adjuvant and compared to a 400 ug dose of non-adjuvanted post-fusion F in RSV seropositive adult cows. There were three groups, 9 animals per group for the DS-Cav1-F 60 ug and 400 ug vaccine doses, and 4 animals for the post-F 400 ug vaccine. Both vaccine doses of DS-Cav1 F induced significant rises in serum neutralizing antibodies that were directed mostly to the pre-fusion antigenic sites, while the post-fusion F was poorly immunogenic. A dose response was not observed between the 60 and 400 ug doses of DS-Cav1 F vaccine. Peak responses were observed at 14 days post-vaccination with a serum half-life of 46 to 37 days.

The novel aspect of this study was the evaluation of an RSV F vaccine in RSV seropositive cows. The RSV seropositive cows served as a model to determine the immunogenicity of an RSV F vaccine in an RSV experienced host. The authors demonstrated convincingly the relatedness of the bovine RSV F to the human RSV F and the existence of the major pre-fusion & post-fusion antigenic sites. The second relevant finding was the observation that the pre-fusion F and not the post-fusion F nonadjuvanted vaccine was associated with a significant increase in serum neutralizing antibody response directed primarily to pre-fusion antigenic sites. A concern of the study is the quality of the post-fusion F vaccine used and the low numbers of cows used in this group. The post-fusion F did not induce functional (neutralizing antibodies) antibodies and data was not presented if binding antibodies (ELISA IgG antiF) were induced post-vaccination. It is unclear if the superiority of the prefusion F vaccine observed in this study is due to the quality of the response induced by the pre-fusion F vaccine or because the post-fusion F vaccine was of poor quality (e.g aggregated, degraded, or unstable).

A second concern is that neutralizing antibody responses against RSV/B was not included. It is important to include the RSV/B neutralizing antibody data because in any given RSV season, RSV/B viruses will contribute to the RSV attach rate and in some seasons, it will be the dominant virus subgroup.

Lastly, it is unclear if the RSV seropositive cows were seropositive because of natural infection with bovine RSV or vaccination with a bovine RSV vaccine. In adults (humans), the RSV seropositive state will be from natural infection. Please comment how the data might be affected if the RSV seropositive state was caused by bovine RSV vaccine rather than natural infection.

Other comments.

Introduction

Pg 4, line 72. Please verify that reference 24 is the appropriate reference. The article by J Mousa et al (A novel pre-fusion conformation-specific neutralizing epitope on the RSV F protein in nature Microbiology 2017) seems more appropriate.

Figures 4 & 5. In most of the RSV seropositive cows, the prefusion F protein was able to

absorb out a major percentage of the serum neutralizing activity after vaccination with DS-Cav1 F. However, antigenic site Ø and site II antibodies were comparably induced with the 400 ug dose (data not available for the 60 ug dose). Also, antibodies to site Ø is considered the major component of the neutralizing antibody activity, however it was not observed in the cows prior to vaccination. In the discussion section, it would be useful to further discuss antibodies to other antigenic sites that might be the major contributor(s) to the neutralizing antibody response.

There is good data in both children and adults that post-vaccination with a post-fusion F protein vaccine the neutralizing antibody decay is about 0.2log every 30 days, which is much slower than the 36-46 day half-life (or 1 log reduction every 36-46 days) experienced by the DS-Cav1 F vaccinated cows. Is this unique to the DS-Cav1 vaccine or other pre-fusion F vaccines? Is this also true for serum neutralizing antibody decay following experimental infection with human RSV in cows? Please discuss in the discussion section.

Discussion.
Pg 20, line 367-368. The statement that DS-Cav1 F induced a poor booster response to neutralizing antibody binding of sites II and IV should be revised because it is not supported by the data presented in figure 5 for site II binding and no data was presented for site IV.

Reviewer #2 (Remarks to the Author):

The study by Steff and colleagues analyzes the ability of hRSV pre-F antigen to boost pre-existing bovine RSV (bRSV) nAb that are cross-reactive with hRSV. This manuscript is fairly well written, clear, concise, and the methods are convincing. Overall, the study demonstrates the potential for RSV pre-F to be an effective vaccine in the setting of maternal immunization, especially compared to post-F, which implies the conformation of the protein differentiates from previous failed subunit boosting vaccines like post-F purified from virus preparations and post-F protein rosettes. This is an important contribution to RSV vaccine development. The study would benefit from a more detailed analysis of anti-hRSV neutralizing activity beyond the old Long strain, and in lieu of any efficacy data. Additional comments are outlined below.

Specific Comments

1. The following sentence in the abstract has no business being the abstract, "Such RSV neutralizing titers would confer 4-5 months protection in babies following maternal immunization." The study does not demonstrate or provide supportive data for this conclusion. In fact, there are no efficacy data in the manuscript. The sentence is an unscholarly pitch.
2. Although the antisera can neutralize Long and the bRSV strain, the investigators should determine titers against recent hRSV A and B strains. Since bRSV is more distantly related to hRSV A than is hRSV B, one might anticipate ample neutralization of hRSV subgroup strains. However, the current study does remind us that the G protein could be a factor. The G protein in recent hRSV A and B strains has evolved, with some duplicated regions, not present in Long strain or bRSV. So if G is, for example, shielding epitopes of F, this may be a confounding problem, which could be studied with a few more nAb titers against hRSV

isolates.

3. It would be more convincing to show, potentially in supplemental figure, biochemical evidence of the integrity of the post-F protein.
4. One line 157, the antigen did not boost pre-existing hRSV responses. Rather, the antigen boosted pre-existing bRSV responses that are cross-reactive with hRSV.
5. The authors should comment more specifically on the lack of immunogenicity in aporox 10% of animals. If it is G, how does that prevent induction of Ab by F?
6. Discussion line 281. Given the results of the FI-RSV trials, and given the poor safety performance of RSV subunit vaccines in pre-clinical RSV models, the risks here are easily perceived, as opposed to an unperceived or unexpected risk. Suggest alternate phrasing.
7. In the Discussion, the authors state boosting was dose-dependent, but in the Results, it was noted that there was not significant difference between the 400 ug and the 60 ug dose. Which is it?

Minor Comments

1. Be consistent: RSV or hRSV.
- 2.

Reply to reviewers' comments (please note that the line numbers mentioned below refer to the revised version of the manuscript).

Reviewer #1 (Remarks to the Author):

The novel aspect of this study was the evaluation of an RSV F vaccine in RSV seropositive cows. The RSV seropositive cows served as a model to determine the immunogenicity of an RSV F vaccine in an RSV experienced host. The authors demonstrated convincingly the relatedness of the bovine RSV F to the human RSV F and the existence of the major pre-fusion & post-fusion antigenic sites. The second relevant finding was the observation that the pre-fusion F and not the post-fusion F non-adjuvanted vaccine was associated with a significant increase in serum neutralizing antibody response directed primarily to pre-fusion antigenic sites.

R: We thank the reviewer for the positive comments on the relevance of our findings.

- 1) A concern of the study is the quality of the post-fusion F vaccine used and the low numbers of cows used in this group. The post-fusion F did not induce functional (neutralizing antibodies) antibodies and data was not presented if binding antibodies (ELISA IgG antiF) were induced post-vaccination. It is unclear if the superiority of the pre-fusion F vaccine observed in this study is due to the quality of the response induced by the pre-fusion F vaccine or because the post-fusion F vaccine was of poor quality (e.g aggregated, degraded, or unstable).

R: We appreciate this reviewer's concern on the quality of the post fusion RSV F used in this study. We and others have previously reported the structure of post-fusion RSV F (Post F) and its biochemical characterization, demonstrating that it is an extremely stable molecule (i.e. even when kept at high temperatures). To satisfy the reviewer request we have added two supplementary figures (Supplementary Figures 1 and 2) summarizing the quality controls of the Post F and DS-Cav1 proteins used in the study. Supplementary Figure 1 shows the biochemical characterization of the molecules used in this study by and UPLC (for integrity, trimerization and lack of aggregation). Supplementary Figure 2 shows the antigenic characterization (i.e. binding to conformation dependent antibodies by ELISA) and mouse immunogenicity data for post-fusion F and Ds-Cav1 pre-fusion F. The in vitro data confirm the good quality of the post-fusion F protein used in the study and the neutralizing antibodies titers obtained in the in vivo study are consistent with those reported previously by us and other groups. Finally, an additional table (Supplementary Table 2) provides the bRSV IgG data for post-fusion F, also showing that some levels of RSV-binding antibodies were elicited by Post F.

In summary these data clearly demonstrate that the Post F used in this study is of very high quality and immunogenic in mice. Therefore, poor quality of the post fusion RSV F molecule can be excluded as potential cause of the lack of boosting in the bRSV infected cows.

We now allude to the characterization of post-fusion F in the Results section (line 148) where the following sentence was added: "The Post F and pre-fusion DS-Cav1 used in this study were fully characterized biochemically, antigenically and also in an in vivo mouse immunogenicity experiment (Supplementary Fig. 1 and Supplementary Fig. 2). Both proteins

were shown to be of high quality and having the same in vitro and in vivo properties as reported in previous studies^{18, 21}.”

- 2) A second concern is that neutralizing antibody responses against RSV/B was not included. It is important to include the RSV/B neutralizing antibody data because in any given RSV season, RSV/B viruses will contribute to the RSV attach rate and in some seasons, it will be the dominant virus subgroup.

R: We agree with the reviewer’s assessment on the contribution of of RSV/B on the overall yearly attack rate of RSV. To address this point we have tested the Day 0 and Day 14 sera of cattle immunized with Post F or Pre F DS-Cav1 in neutralization of RSV/B. These data are now presented in Table S1 and Figure 3B. The new data show similar neutralization potency on the RSV A and B strains.

The following sentence was added to the results section:

Results (line 170): “Overall, hRSV B neutralization response was very similar to that observed with hRSV A (Fig. 3 and Table S1), showing that antibodies raised through vaccination of bRSV-positive animals with a pre-fusion F derived from a hRSV A strain were cross-reactive with another RSV group. A minimal increase in hRSV B neutralization response was observed after vaccination with PostF.”

- 3) Lastly, it is unclear if the RSV seropositive cows were seropositive because of natural infection with bovine RSV or vaccination with a bovine RSV vaccine. In adults (humans), the RSV seropositive state will be from natural infection. Please comment how the data might be affected if the RSV seropositive state was caused by bovine RSV vaccine rather than natural infection.

As mentioned in the manuscript (line 304), the vaccination history of the animals used in this study could not be retraced. In Belgium, the country of origin of the animals, attenuated or inactivated bRSV vaccines are available. Rispoval®, an intra-nasal live-attenuated vaccine is the most commonly administered, during the first months of life only. Given the age of the animals, the high prevalence of bRSV infections in previous years and the non-sterilizing immunity conferred by the vaccine, it is very likely that the animals underwent natural bRSV infections in the interval between potential vaccination and the study conduct.

- 4) Other comments.
 - a) Introduction: Pg 4, line 72. Please verify that reference 24 is the appropriate reference. The article by J Mousa et al (A novel pre-fusion conformation-specific neutralizing epitope on the RSV F protein in nature Microbiology 2017) seems more appropriate.

R: We thank the reviewer for the suggestion. The reference has been modified accordingly.

- b) Figures 4 & 5. In most of the RSV seropositive cows, the prefusion F protein was able to absorb out a major percentage of the serum neutralizing activity after vaccination with DS-Cav1 F. However, antigenic site Ø and site II antibodies were comparably induced

with the 400 ug dose (data not available for the 60 ug dose). Also, antibodies to site Ø is considered the major component of the neutralizing antibody activity, however it was not observed in the cows prior to vaccination. In the discussion section, it would be useful to further discuss antibodies to other antigenic sites that might be the major contributor(s) to the neutralizing antibody response.

The fact that pre-fusion F but not post-fusion F depleted most of the neutralizing titers from the sera of bRSV infected cattle suggests that antibodies binding to site Ø and other pre-fusion specific sites are responsible for most of the RSV neutralization. Although site II and site Ø antibodies were comparably induced by DS-Cav1 vaccination, it is known that site II antibodies are generally less potent than site Ø antibodies; hence the boost of site II antibodies may only marginally contribute to total neutralization. This was clarified in the discussion section (line 380).

The fact that we do not detect pre-existing antibodies competing to site Ø or site II in cattle infected by bRSV F may be due to relatively low sensitivity of the assay. This was clarified in the results section (line 245).

It should be kept in mind that in these competition assays, the addition of site II- or site Ø-binding monoclonal antibodies can also displace antibodies from other nearby epitopes, such as the MPE-8 site or the recently discovered site VIII, respectively. Although these details could be further discussed in the manuscript, we believe that this goes beyond the scope of the manuscript and is already known in the RSV field.

- c) There is good data in both children and adults that post-vaccination with a post-fusion F protein vaccine the neutralizing antibody decay is about 0.2log every 30 days, which is much slower than the 36-46 day half-life (or 1 log reduction every 36-46 days) experienced by the DS-Cav1 F vaccinated cows. Is this unique to the DS-Cav1 vaccine or other pre-fusion F vaccines? Is this also true for serum neutralizing antibody decay following experimental infection with human RSV in cows? Please discuss in the discussion section.

R: We thank the reviewer for this comment. We have reviewed most of the available data regarding hRSV neutralizing antibody half-life in adults and children (after vaccination) or in newborns (passive transfer from maternal antibodies). Depending on the intervention, assay used and target population, half-lives appear to vary in these studies, ranging anywhere from 36 to 76 days. The Discussion (line 361) has been updated to reflect this and we used conservative estimates to calculate a potential duration of protection of maternal immunization with the DS-Cav1 vaccine.

The discussion now reads “The antibody half-lives calculated in this study, 46 and 37 days for the 400 and 60 µg doses of DS-Cav1, respectively, are consistent with studies reporting RSV neutralizing antibody half-lives of 36 to 38 days in newborns following RSV infection^{43, 44} or vaccination⁴⁵ of their mothers. Other studies however have shown much longer antibody persistence in newborns⁹ or vaccinated children⁴⁶. We tried to estimate the duration of protection that would be conferred by a 14-fold increase in

neutralization titers from baseline, as seen 14 days after vaccination with high dose DS-Cav1.”

- 5) Discussion. Pg 20, line 367-368. The statement that DS-Cav1 F induced a poor booster response to neutralizing antibody binding of sites II and IV should be revised because it is not supported by the data presented in figure 5 for site II binding and no data was presented for site IV.

R: Pre-fusion F Ds-Cav-1 boosted antibodies competing with Palivizumab (site II) but these antibodies are known to have a lower neutralizing potential, which is confirmed by the depletion experiment with post-fusion F, which contains both site II and IV, where neutralizing titers were not affected. We have revised the text to clarify that we are referring to neutralizing potential and not antibodies in general (which are clearly boosted by pre-fusion F). It now reads (line 378): “We also note that the contribution of site II and site IV epitopes, shared by pre- and post-fusion F, to the overall neutralizing antibody response is probably minor, as addition of post-fusion F to the sera only marginally affected neutralizing titers”.

Reviewer #2 (Remarks to the Author):

The study by Steff and colleagues analyzes the ability of hRSV pre-F antigen to boost pre-existing bovine RSV (bRSV) nAb that are cross-reactive with hRSV. This manuscript is fairly well written, clear, concise, and the methods are convincing. Overall, the study demonstrates the potential for RSV pre-F to be an effective vaccine in the setting of maternal immunization, especially compared to post-F, which implies the conformation of the protein differentiates from previous failed subunit boosting vaccines like post-F purified from virus preparations and post-F protein rosettes. This is an important contribution to RSV vaccine development. The study would benefit from a more detailed analysis of anti-hRSV neutralizing activity beyond the old Long strain, and in lieu of any efficacy data. Additional comments are outlined below.

Specific Comments

- 1) The following sentence in the abstract has no business being the abstract, “Such RSV neutralizing titers would confer 4-5 months protection in babies following maternal immunization.” The study does not demonstrate or provide supportive data for this conclusion. In fact, there are no efficacy data in the manuscript. The sentence is an unscholarly pitch.

R: We agree with the reviewer’s comments and have removed the sentence from the abstract.

- 2) Although the antisera can neutralize Long and the bRSV strain, the investigators should determine titers against recent hRSV A and B strains. Since bRSV is more distantly related to hRSV A than is hRSV B, one might anticipate ample neutralization of hRSV subgroup strains. However, the current study does remind us that the G protein could be a factor. The G protein in recent hRSV A and B strains has evolved, with some duplicated regions, not present in Long strain or bRSV. So if G is, for example, shielding epitopes of F, this may be a confounding problem, which could be studied with a few more nAb titers against hRSV isolates.

R: We agree with the reviewer’s assessment on the contribution of of RSV/B on the overall yearly attack rate of RSV. To address this point we have tested the Day 0 and Day 14 sera of cattle immunized with Post F or Pre F DS-Cav1 in neutralization of RSV/B. These data are now presented in Table S1 and Figure 3B. The new data show similar neutralization potency on the RSV A and B groups. We were unfortunately unable to access other RSV isolates to perform additional neutralization assay. However, given the cross-reactivity observed between hRSV A, hRSV B and bRSV, it is probable that the F protein used to immunize animals is contributing to the vast majority of the neutralizing antibody response.

The following sentence was added to the results section:

Results (line 170): “Overall, hRSV B neutralization response was very similar to that observed with hRSV A (Fig. 3 and Table S1), showing that antibodies raised through vaccination of bRSV-positive animals with a pre-fusion F derived from a hRSV A strain were

cross-reactive with another RSV group. A minimal increase in hRSV B neutralization response was observed after vaccination with PostF.”

- 3) It would be more convincing to show, potentially in supplemental figure, biochemical evidence of the integrity of the post-F protein.

R: We appreciate this reviewer’s concern on the quality of the post fusion RSV F used in this study. We and others have previously reported the structure of post-fusion RSV F (Post F) and its biochemical characterization, demonstrating that it is an extremely stable molecule (i.e. even when kept at high temperatures). To satisfy the reviewer request we have added two supplementary figures (Supplementary Figures 1 and 2) summarizing the quality control of the Post F protein used in the study. Supplementary Figure 1 shows the biochemical characterization of the Post F molecule used in this study by and UPLC (for integrity, trimerization and lack of aggregation). Supplementary Figure 2 shows the antigenic characterization (i.e. binding to conformation dependent antibodies by ELISA) and mouse immunogenicity data for post-fusion F and Ds-Cav1 pre-fusion F. The in vitro data confirm the good quality of the post-fusion F protein used in the study and the neutralizing antibodies titers obtained in the in vivo study are consistent with those reported previously by us and other groups. Finally, an additional table (Supplementary Table 2) provides the bRSV IgG data for post-fusion F, also showing that some levels of RSV-binding antibodies were elicited by Post F.

In summary these data clearly demonstrate that the Post F used in this study is of very high quality and immunogenic in mice. Therefore, poor quality of the post fusion RSV F molecule can be excluded as potential cause of the lack of boosting in the bRSV infected cows.

We now allude to the characterization of post-fusion F in the Results section (line 148) where the following sentence was added: “The Post F and pre-fusion DS-Cav1 used in this study were fully characterized biochemically, antigenically and also in an in vivo mouse immunogenicity experiment (Supplementary Fig. 1 and Supplementary Fig. 2). Both proteins were shown to be of high quality and having the same in vitro and in vivo properties as reported in previous studies^{18,21}.”

- 4) One line 157, the antigen did not boost pre-existing hRSV responses. Rather, the antigen boosted pre-existing bRSV responses that are cross-reactive with hRSV.

R: We agree with the reviewer’s comments and have modified the sentence accordingly.

- 5) The authors should comment more specifically on the lack of immunogenicity in approx 10% of animals. If it is G, how does that prevent induction of Ab by F?

R: It was not totally clear to us how the “lack of immunogenicity in approx. 10% of animals” was calculated. Among the 22 animals vaccinated in this study, one animal did not respond to the vaccine in terms of anti-RSV binding antibodies or RSV neutralizing antibodies (non-response rate of 4.5%). Based on our experience with this model, this is a very unusual observation and we therefore considered this animal as an outlier, not including it in subsequent calculations.

- 6) Discussion line 281. Given the results of the FI-RSV trials, and given the poor safety performance of RSV subunit vaccines in pre-clinical RSV models, the risks here are easily perceived, as opposed to an unperceived or unexpected risk. Suggest alternate phrasing.

We removed the word “perceived”, which in our opinion clarifies the sentence.

- 7) In the Discussion, the authors state boosting was dose-dependent, but in the Results, it was noted that there was not significant difference between the 400 ug and the 60 ug dose. Which is it?

We have modified the text to indicate that although the Day 14 to Day 0 ratio was greater in the high dose group, the difference in boosting between the 400 and 60 µg dose groups did not reach statistical significance (line 344).

- 8) Minor Comments: Be consistent: RSV or hRSV.

Thank you for pointing this out. This has now been corrected. We have kept the term RSV (without specifying human or bovine) when RSV was used as a generic term.

REVIEWERS' COMMENTS:

Reviewer #2 (Remarks to the Author):

The issues and questions of were adequately and thoroughly addressed. This reviewer has no additional questions or concerns.